# Hydrolyzed Conchiolin Protein (HCP) Extracted from Pearls Antagonizes both ET-1 and *α*-MSH for Skin Whitening

**DOI:** 10.3390/ijms24087471

**Published:** 2023-04-18

**Authors:** Shan Yang, Zhekun Wang, Yunwei Hu, Kaile Zong, Xingjiang Zhang, Hui Ke, Pan Wang, Yuyo Go, Xi Hui Felicia Chan, Jianxin Wu, Qing Huang

**Affiliations:** 1Skin Health and Cosmetic Development & Evaluation Laboratory, China Pharmaceutical University, Nanjing 211198, China; 2Royal Victoria Hospital, 274 Grosvenor Rd, Belfast BT12 6BA, UK; 3Acute Mental Health Inpatient Centre, Belfast BT9 7YG, UK

**Keywords:** HCP (hydrolyzed conchiolin protein), Pearl, whitening effects, MNT-1 melanocytes

## Abstract

Pearl powder is a famous traditional Chinese medicine that has a long history in treating palpitations, insomnia, convulsions, epilepsy, ulcers, and skin lightining. Recently, several studies have demonstrated the effects of pearl extracts on protection of ultraviolet A (UVA) induced irritation on human skin fibroblasts and inhibition of melanin genesis on B16F10 *mouse* melanoma cells. To further explore the effect we focused on the whitening efficacy of pearl hydrolyzed conchiolin protein (HCP) on *human* melanoma MNT-1 cells under the irritation of alpha-melanocyte-stimulating hormone (*α*-MSH) or endothelin 1 (ET-1) to evaluate the intracellular tyrosinase and melanin contents, as well as the expression levels of tyrosinase (*TYR*), tyrosinase related protein 1 (*TRP-1*), and dopachrome tautomerase (*DCT*) genes and related proteins. We found that HCP could decrease the intracellular melanin content by reducing the activity of intracellular tyrosinase and inhibiting the expression of TYR, TRP-1, DCT genes and proteins. At the same time, the effect of HCP on melanosome transfer effect was also investigated in the co-culture system of immortalized *human* keratinocyte HaCaT cells with MNT-1. The result indicated that HCP could promote the transfer of melanosomes in MNT-1 melanocytes to HaCaT cells, which might accelerate the skin whitening process by quickly transferring and metabolizing melanosomes during keratinocyte differentiation. Further study is needed to explore the mechanism of melanosome transfer with depigmentation.

## 1. Introduction

The Melanocytes, located in the stratum basal layer of the epidermis, play a crucial role in preventing the photocarcinogenic effects of solar ultraviolet (UV) [1,2]. With the stimulus of UV rays, keratinocytes produce hormones, such as alpha-melanocyte-stimulating hormone (*α*-MSH), endothelin 1 (ET-1) and adrenocorticotropic hormone, which interact with the receptors on melanocyte to induce synthesis of melanin and trigger the formation melanocyte dendrite spontaneously [3,4]. Mature melanosome containing melanin pigment transports to the keratinocytes through arborized dendrites [5]. Melanin arranged annularly around the nucleus in the keratinocytes scavenges free radicals and protects the skin from the damages of UV light on DNA [6]. 

Alpha-melanocyte-stimulating hormone (*α*-MSH) is a small peptide cleaved from the precursor pro-opiomelanocortin (POMC) [7]. *α*-MSH has been shown to act via activation of the downstream target of protein kinase A (PKA) and melanocyte inducing transcription factor (MITF). MITF has a pivotal role in the regulation of melanogenesis and transport related genes including, but not limited to, tyrosinase (*TYR*), tyrosinase related Protein 1 (*TRP1*), dopachrome tautomerase (*DCT*), premelanosome protein (*PMEL17*), and *Rab27a* [8,9,10]. Melanogenesis commences with the conversion of *L*-tyrosine to *L*-dihydroxyphenylalanine (*L*-DOPA) catalyzed by tyrosinase key rate-limiting enzyme [11,12].

Meanwhile, binding endothelin-1 (ET-1), a vasoactive peptide, with the ET-B receptor on melanocytes could improve melanogenesis via the phosphorylation of the downstream targets of protein kinase C (PKC) and melanocyte inducing transcription factor (MITF) [13].

Although there are differences in binding receptors and downstream pathways, both *α*-MSH and ET-1 eventually bind to melanocytes to promote melanin production and transport by upregulating MITF and P53 [2].

Excessive melanin production is associated with many manifestations including but not limited to senile lentigo, hyperpigmentation, melasma and solar lentigo [1,14]. Whitening is not simply to whiten the skin, but to prevent and eliminate excessive pigmentation. Excessive pigmentation, especially in Asian yellow-skinned people, can directly lead to obvious skin color changes such as dullness, yellowing, and opacity. It is estimated that approximately 15% of the world population have invested in skin whitening agents, with Asia being the vast majority of it [15]. At the same time, preventing skin hyperpigmentation and eliminating melanin formation is a time-consuming process that can cause psychological distress to consumers, and the use of whitening agents can help with this process. However, some of skin-whitening agents such as rhododendrol and hydroquinone cause significant side-effects such as leukoderma and erythema [16,17]. The development of safe and effective whitening agents is in line with market demand.

Pearl is a famous traditional Chinese medicine recorded in the Pharmacopoeia of the People’s Republic of China with soothing eyesight, detoxification, skin moisturizing and flighting effects. Pearl powder has been used in traditional Chinese medicine prescription to achieve the efficacy of skin whitening [18]. The main chemical composition of pearl is about 82–86% calcium carbonate, 10–14% protein, and 2–4% water. It has been demonstrated that soluble pearl extract can scavenge reactive oxygen, free radicals and inhibit the depigmenting activity of tyrosinase [17,19]. Jing Wang MD et al. found anti-melanogenic effects of soluble pearl extract (SPE) in B16F10 melanoma cells [20]. In this report, we explored the effects of hydrolyzed conchiolin protein (HCP) obtained from pearl powder on melanogenesis in MNT-1 human melanoma cells.

## 2. Results

### 2.1. Effects of α-MSH and ET-1 on MNT-1 Cells

The MNT-1 melanoma cells were treated with different concentrations of α-MSH and ET-1 for 48 h, and the proliferation of the cells was evaluated by MTT assay respectively (Figure 1). Notably, the treatment with α-MSH from 0.03 up to 1 *μg*/mL promoted proliferation of MNT-1 melanoma cells significantly in a concentration dependent manner compared with the control (*p* < 0.001). The treatment with ET-1 from 0.33 up to 5 μg/mL had no influence on the proliferation of MNT-1. However, when the concentration was increased to 10 μg/mL, ET-1 promoted the proliferation of MNT-1 melanoma cells (*p* < 0.001). 

α-MSH at 0.01 and 0.1 μg/mL could trigger MNT-1 to produce more melanin in a concentration dependent manner compared with the control, while ET-1 induced significantly more melanin production of MNT-1 cells at 1 μg/mL. Thus 0.1 μg/mL of α-MSH and 1 μg/mL of ET-1 were used in subsequent experiments.

### 2.2. Effects of HCP on the Viabilities of MNT-1 Melanoma and HaCaT Cells

The noncytotoxic concentration range of HCP on MNT-1 melanoma cells and HaCaT human keratinocytes were identified by MTT assay. The results showed that HCP was nontoxic over the range of 0–2.5 mg/mL and enhanced MNT-1 proliferation at 2.5 mg/mL. HCP reduced the viability of MNT-1 by 14.7% at 5 mg/mL and by 74.0% at 10 mg/mL (Figure 2a). HCP was nontoxic to HaCaT over the range of 0–1.25 mg/mL and induced significant cytotoxicity at 2.5, 5, 10 mg/mL (Figure 2b).

### 2.3. Effects of HCP on the Melanin Content in MNT-1

We evaluated the anti-melanogenic effects of HCP on MNT-1 by measuring the intracellular melanin content. Results showed when treated with 0.1 μg/mL of α-MSH, MNT-1 significantly upregulated the intracellular melanin content to 109.32% compared with the no α-MSH treatment group. While after adding HCP, MNT-1 reduced the α-MSH induced melanin content in a concentration dependent manner by 5.41%, 34.42% and 50.73% at 0.2, 0.45 and 0.7 mg/mL, respectively. Meanwhile, kojic acid at 0.2 mg/mL decreased the α-MSH induced melanin content by 48.54% (Figure 3a). On the other hand, MNT-1 treated with 1 μg/mL of ET-1 significantly upregulated the intracellular melanin contents to 111.66% compared with the no ET-1 treatment group. HCP showed anti-melanogenesis by significantly decreasing ET-1 induced MNT-1 intracellular melanin content by 49.05%, 29.59% and 40.12% at the concentration of 0.2, 0.45 and 0.7 mg/mL, respectively. While, 0.2 mg/mL α-arbutin decreased the ET-1 induced melanin content by 11.51% (Figure 3b).

### 2.4. Effects of HCP on the Tyrosinase Activity in MNT-1

To further explore the inhibitory mechanism of HCP on the synthesis of melanin, we analyzed the intracellular tyrosinase inhibition effort of HCP on MNT-1 by the L-dopa oxidation method. As is shown in Figure 4, with the treatment of ET-1, the tyrosinase activity significantly peaked by 7.5% when compared with that in the untreated cells. While HCP markedly diminished the ET-1-induced intracellular tyrosinase activity by 12.6, 16.4 and 15.2% at 0.2, 0.45, 0.7 mg/mL, respectively. As a positive control, 0.2 mg/mL α-arbutin decreased the ET-1-induced intracellular tyrosinase activity by 11.70%.

As shown in the Figure 4, with the treatment of α-MSH, tyrosinase activity in MNT-1 significantly peaked by 12.60% when compared with that in the untreated cells, which was significantly inhibited by following the treatment with HCP in a concentration dependent manner (6.0%, 7.5% and 25.9% at 0.2, 0.45, 0.7 mg/mL, respectively, Figure 4a). As a comparison, 0.2 mg/mL kojic acid decreased tyrosinase activity by 34.17%.

### 2.5. Effects of HCP on TYR, TRP-1 and DCT mRNA Levels in MNT-1 Melanoma Cells

To investigate the mechanism of HCP altering the effects of ET-1 and α-MSH induced melanin generation and tyrosinase activity, the mRNA levels of *TYR*, *TRP-1* and *DCT* were measured in MNT-1 melanoma cells after being treated with ET-1 or α-MSH (Figure 5). We found that with ET-1 and α-MSH treatments at 1 μg/mL and 0.1 μg/mL, respectively, the mRNA levels of *TYR*, *TRP-1* and *DCT* in MNT-1 cells increased significantly compared with the control groups. However, together with the HCP treatment, the mRNA expression of *TYR* and *TRP-1* and *DCT* were effectively inhibited at all concentration levels both in the ET-1 and α-MSH treated MNT-1 cells. As the positive controls, kojic acid and α-arbutin also decreased the expression of those genes.

### 2.6. Effects of HCP on TYR, TRP-1 and DCT Protein Levels in MNT-1 Melanoma Cells

The protein expressions of TYR, TRP-1 and DCT in MNT-1 melanoma cells were detected by Western-blot assays (Figure 6). With the treatment of ET-1 and α-MSH respectively, the protein levels of TYR, TRP-1 and DCT in MNT-1 were significantly increased when compared with the control groups. As expected, HCP could significantly inhibit the TYR, TRP-1 and DCT protein expressions both in the absence and presence of α-MSH or ET-1. Kojic acid and α-arbutin also decreased the expression of protein as the positive controls. 

### 2.7. Effects of HCP on Melanosome Transfer in the Co-Culture of Melanoma and Keratinocytes

Transfer of melanin occurs between melanocytes and keratinocytes. Keratinocytes produce secretion factors that regulate the proliferation, differentiation and melanogenesis of melanocytes [21]. In addition, activation of protease-activated receptor-2 (PAR-2) on keratinocytes and initiation of intracellular calcium signaling promote the transfer of melanin [22,23]. Adopting cultures consisting purely of melanocytes ignores the influence of keratinocytes. The use of keratinocyte-melanocyte co-culture models can better simulate the physiological environment and reflect the effect of HCP on melanin transfer.

After co-culture of MNT-1 with HaCaT cells for 48 h, the efficiency of melanosome transfer was quantitatively measured using flow cytometry. As shown in Figure 7B, the number of keratinocyte cells containing melanosomes are shown in the upper right quadrant (Q2). Co-culture for 48 h in the presence of HCP resulted in promotion of the ratio of keratinocytes with melanosomes (21.20%, 18.10% 7.77% at 0.2, 0.45 and 0.7 mg/mL, respectively). To further verify the result of flow cytometry, we conducted an immunofluorescent double-staining experiment (Figure 7a). At 0.7 mg/mL, HCP significantly increased the colocalization correlation coefficient of TRP-1 and pan cytokeratin by 20.6%. While as a comparison, nicotinamide at 0.2 mg/mL decreased the colocalization correlation coefficient of TRP-1 and pan cytokeratin by 49.76%.

## 3. Discussion

As known, *α*-MSH and ET-1 synthesize melanin and regulate pigmentation through the binding of MC1R and EDNRB receptors. Although the influencing factors and signaling pathways are different, they all converge on MITF and P53 and affect the downstream pathways [10,24]. For melanin synthesis, activation of MITF leads to the upregulation of TYR, TRP-1 and DCT, where TYR is the rate-limiting/key enzyme in melanin production [10,25]. TRP-1 acts as an oxidase of 5,6-dihydroxyindole-2-carboxylic acid [DHICA] in the biosynthetic conversion of tyrosine to eumelanin [26], while DCT isomerizes dopachrome to DHICA rather than DHI [27]. Furthermore, ET-1 may indirectly affect melanocytes in the skin by up-regulating the mRNA level of *MC1R* and enhancing the level of melanocyte response to *α*-MSH [28].

As mentioned before, several studies on pearl powder and extracts have demonstrated their whitening effect [17,18,19,20]. This is the first time the purified pearl extract HCP on MNT-1 human melanoma cells has been studied to observe the whitening effect. 

Pearl has been used in medicine for more than one thousand years in China, and its safety has been verified. HCP was nontoxic over the range of 0–2.5 mg/mL and 0–1.25 mg/mL to MNT-1 and HaCaT cells and it can exert a significant whitening effect at 0.2 mg/mL. The extract is derived from nature and meets consumers’ expectations for whitening ingredients. Moreover, HCP has exhibited low adverse effects and is a real nature sourcing ingredient in comparison to frequently used actives in cosmetic products such as arbutin, kojic acid and hydroquinone which may be prohibited in some countries.

In this study, we confirmed the increasing tyrosinase activity and melanin content in MNT-1 cells via the treatment with either ET-1 at 1 μg/mL or *α*-MSH at 0.1 μg/mL. However, HCP could alter the effects of ET-1 and *α*-MSH on tyrosinase activity and melanin content in MNT-1 cells. In addition, corresponding to the tyrosinase activity and the melanin content decrease, the mRNA and protein levels of TYR, TRP-1 and DCT were suppressed as well, when compared with HCP untreated cells. In the MNT-1 and HaCaT coculture study, we observed with flow cytometry and immunofluorescent double-staining techniques that HPC could accelerate the mistake transfer from melanocyte to keratinocyte. Although the mechanism of melanosome transferring from melanocyte to keratinocyte is still unclear, it is a quite important step to maintain and protect the healthiness of cells. Melanin deposited in the basal layer has melanogenic properties to melanocytes and inhibition on keratinocytes proliferation [29,30]. Meanwhile, the content of melanin in the epidermis can be decreased by excretion and reduction reactions to achieve the aim of whitening. For safe whitening of skin, the focus should be on promoting the metabolization and proliferation of keratinocytes rather than preventing melanosome transferring. Further study is needed to explore the mechanism of HCP of melanosomes transfer with depigmentation. 

## 4. Materials and Methods

### 4.1. Hydrolyzed Conchiolin Protein (HCP) and Preparation of HCP Solution

HCP was provided by OSM Biology Co., Ltd. (Huzhou, China), and prepared by following steps. First, pearl powder was dissolved in lactic acid, and the pH of the mixture was adjusted to 7.0 with PBS. Enzymatic hydrolysis was initiated with addition of protease, and the extract was filtered with 10 μm qualitative filter paper. The supernatant was lyophilized to obtain hydrolyzed conchiolin protein (HCP). HCP solution was prepared by dissolving HCP in PBS (pH = 7.4) and filtrate with 0.22 μm membrane filter.

### 4.2. Analysis of the Hydrolyzed Conchiolin Protein (HCP)

FTIR has been used to detect characteristic absorption protein bands of HCP, amide I (C=O bond, ca. 1630 cm^−1^), amide II (N-H bond, ca. 1553 cm^−1^) and amide III (C-N band, ca. 1418 cm^−1^). The Lowry Method (Lowry Method Protein Kit, Sangong, Shanghai, China) was used to determine the peptide content of the HCP. Following the instruction, a standard curve was established (Y = 0.05521 + 0.85891X), and the protein content of HCP was obtained 42.95%.

### 4.3. Cell Culture

MNT-1 melanoma cells and Human keratinocytes (HaCaT) were obtained from Zhejiang Meisen Cell Technology Co., Ltd. (Hangzhou, China). MNT-1 melanoma cells were cultured in DMEM supplemented with 20% Fetal Bovine Serum, 10% AIM-V, 1% NEAA and antibiotic mixture. HaCaT were cultured in DMEM supplemented with 10% Fetal Bovine Serum and antibiotic mixture. All cells were cultured in a humidified incubator with 95% air–5% CO_2_ at 37 °C.

Cultured keratinocytes and melanocytes were collected by trypsin/EDTA and seeded in six-well plates at the ratio of 2:1. The culture media was composed of HaCaT medium and MNT-1 medium at the ratio of 2:1.

### 4.4. Cell Viability Test

MTT test was used to investigate the cell viability of HCP on cells. MNT-1, HaCaT and melanoma cells were seeded at a density of 5 × 10^4^ cells/mL in 96-well plates, respectively. After 60 h incubation, HCP with different concentrations was added to the wells then incubated for another 48 h. At the end of exposure, the supernatant was aspirated. The cells were treated with MTT (0.5 mg/mL) solution for 4 h at 37 °C, and formazan was dissolved with dimethyl sulfoxide (DMSO). Absorbance was measured at 490/570 nm using SpectraMax 190 plate reader (Molecular Devices Corporation, San Jose, CA, USA) and the results were expressed as a percentage relative to the control.

### 4.5. Measurement of Intracelluar Melanin Content

MNT-1 melanoma cells (1.5× 10^5^ cells/mL) were cultured in 12-well plates and cultured for 60 h. They were treated with the various concentrations of HCP in the presence or absence of 0.1 μg/mL α-MSH or 1 μg/mL ET-1 for 48 h in 2% FBS. At the end of treatment, the cells were detached with Trypsin-EDTA solution, washed with ice-cold PBS and dissolved in 1 M NaOH containing 10% DMSO at 80 °C for 120 min. The absorbance of the lysates was read at 405 nm and the protein content was estimated by BCA assay. The melanin levels of groups were normalized to total protein then expressed as a percentage of the untreated control [13,20].

### 4.6. Intracellular Tyrosinase Activity Assay

MNT-1 melanoma cells (1.6 × 10^5^ cells/mL) were cultured in 12-well plates and cultured for 60 h. They were treated with the various concentrations of HCP in the presence or absence of 0.1 μg/mL α-MSH or 1 μg/mL of ET-1 for 48 h in 2% FBS. Tyrosinase solution was obtained from supernatant of MNT-1 melanoma cells treated with RIPA Lysis Buffer (Solarbio, Beijing, China) and total protein was quantified by Bicinchoninic Acid Assay kit (BCA) (Beyotime, Shanghai, China). An equal amount of protein was mixed with 100 μL of 10 mM L-dihydroxyphenylalanine (L-DOPA) solution (Sigma-Aldrich, St. Louis, MO, USA) at 37 °C for 15 min, and the absorbance of the solution was read at 475 nm. The normalized values were then expressed as a percentage of the untreated control [13,31,32].

### 4.7. Quantitative Real-Time PCR (RT-qPCR) Analysis

Total RNA was extracted from MNT-1 cells using RNA-easy Isolation Reagent. To quantify changes in mRNA levels in MNT-1 cells, 1 μg of total RNA was converted into cDNA with HiScript® III RT SuperMix (Vazyme, Nanjing, China) according to the manufacturer’s protocols. A BIOER LineGene 9600 Plus PCR instrument (Hangzhou, China) and ChamQ SYBR qPCR Master Mix (Low ROX Premixed) (Vazyme, Nanjing, China) were used to quantify the target mRNA level, including TYR, TRP-1 and DCT (Table 1) [33].

### 4.8. Western Blot Analysis

After the indicated treatment, MNT-1 cells were incubated in iced-cold RIPA lysis buffer for 30 min. The concentration of the protein extracts was determined with Bicinchoninic Acid Assay kit (BCA) (Beyotime, Shanghai, China). The proteins were then separated by 10% SDS-PAGE, and the resultant bands electro-transferred onto a Polyvinylidene Fluoride (PVDF) membrane. The membrane was then blocked with QuickBlock™ Blocking Buffer (Beyotime, Shanghai, China) and incubated with antibodies against TYR, TRP-1, DCT and *β*-actin (Santa Cruz, Dallas, TX, USA) at 4 °C for 12 h. Next, the membrane was incubated with the secondary antibody (HRP-labeled Goat Anti-Mouse IgG(H + L)) (Beyotime, Shanghai, China) for 2 h at room temperature. The protein bands were detected using the enhanced chemiluminescence (ECL). The strip scanning analysis was performed using an Image J analysis system. *β*-Actin was used as the internal reference in the analysis of protein expression. Samples were run as independent replicates repeated thrice for each group [29,32].

### 4.9. Flow Cytometry Analysis

Quantitative assessments for melanosome transfer in MC-KC co-culture. Cells in co-culture were treated with the indicated compounds for 48 h. To quantitatively assay melanosome transfer. The co-cultured cells were harvested and washed with cold PBS, fixed in 4% paraformaldehyde for 10 min, washed with PBS containing 0.1% Triton-X100 for 5 min. MCs were immunostained with anti-TRP1 antibody (ab178676, Abcam, Cambridge, UK) and goat anti-mouse IgG H&L/PE secondary antibody (bs-0296G-PE, Bioss, Beijing, China) and KCs were incubated with anti-pan cytokeratin antibody conjugated with alexa fluor 488 (ab277270, Abcam, Cambridge, UK). Stained cells were analysed by flow cytometry. A total of 10 000 events were collected on the CytoFLEX Flow Cytometer (Beckman coulter, Brea, CA, USA). The ratio of percentage of cells positive for both cytokeratin and tyrosinase to that of cells positive for cytokeratin only represents melanosome transfer efficacy [30,34,35].

### 4.10. Immunofluorescence Staining

The co-cultured cells were washed with ice-cold PBS, fixed in 4% paraformaldehyde for 10 min, and washed with PBS containing 0.25% Triton-X100 for 10 min to permeabilize the membrane. Add blocking buffer (Beyotime, Shanghai, China) for 10 min at 37 ℃. Then MCs were immunostained with anti-TRP1 antibody (ab178676, Abcam, Cambridge, UK) and goat anti-mouse IgG H&L/PE secondary antibody (bs-0296G-PE, Bioss, Beijing, China) and KCs were incubated with anti-pan cytokeratin antibody conjugated with alexa fluor 488 (ab277270, Abcam, Cambridge, UK). Nucleus were stained with DAPI. Images were obtained with an Inverted fluorescence microscope (Mshot, Guangzhou, China) and dealt with Image pro plus [36].

## 5. Conclusions

HCP inhibits the activity of tyrosinase and decreases the amount of proteins related with melanogenesis by antagonizing the effect *α*-MSH- and ET-1-induced on MNT-1 human melanoma cells and promoting the transfer to keratinocytes. Results demonstrate the potential of HCP as a safe and effective skin whitening agent.

## Figures and Tables

**Figure 1 ijms-24-07471-f001:**
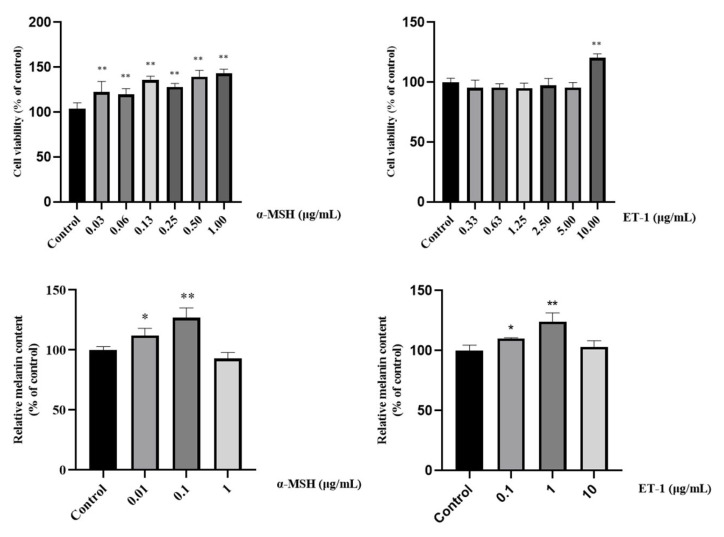
Melanoma cells (MNT-1) viability and melanin content in the presence of different concentrations of Endothelin-1 (ET-1) and alpha-melanocyte-stimulating hormone (*α*-MSH) (* *p* < 0.05; ** *p* < 0.01 vs. control. *t* test, *n* ≥ 3).

**Figure 2 ijms-24-07471-f002:**
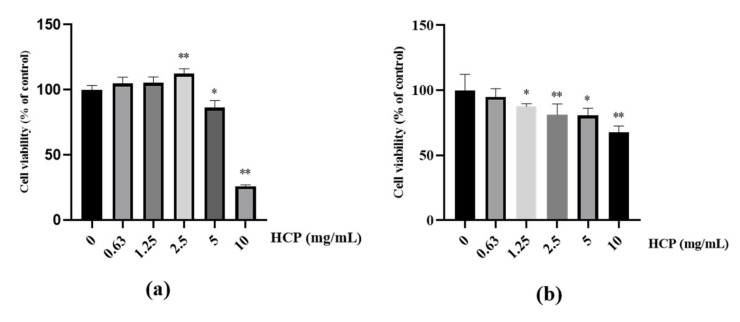
(**a**) Melanoma cells (MNT-1) viability in the presence of different concentrations of hydrolyzed conchiolin protein (HCP) measured by MTT assay (* *p* < 0.05; ** *p* < 0.01 vs. control. *t* test); (**b**) Human keratinocyte (HaCaT) viability in the presence of different concentrations of HCP measured by MTT assay (* *p* < 0.05; ** *p* < 0.01 vs. control. *t* test); all data are mean ± SD of at least triplicates.

**Figure 3 ijms-24-07471-f003:**
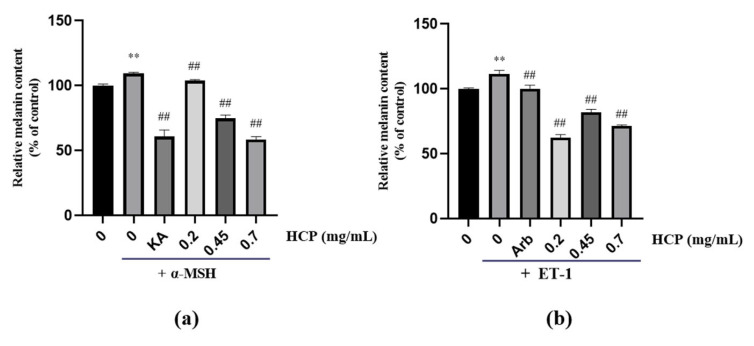
(**a**) HCP inhibits *α*-MSH-induced melanogenesis in MNT-1 melanoma cells. Kojic acid (KA, 0.2 mg/mL) was used as a positive control. (**b**) HCP inhibits ET-1-induced melanogenesis in MNT-1 melanoma cells. α-arbutin (Arb, 0.2 mg/mL) was used as a positive control. All data are expressed as mean ± SD (*n* ≥ 3) (** *p* < 0.01 vs. the untreated control group; ^##^
*p* < 0.01 vs. α-MSH-treated group or ET-1-treated group).

**Figure 4 ijms-24-07471-f004:**
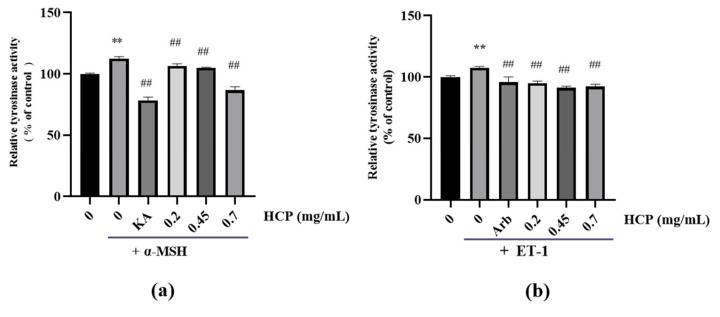
(**a**) HCP inhibits the intracellular tyrosinase of α-MSH-treated MNT-1 melanoma cells. (**b**) HCP inhibits the intracellular tyrosinase of ET-1-treated MNT-1 melanoma cells. All data are expressed as mean ± SD (*n* ≥ 3) (** *p* < 0.01 vs. the untreated control group; ^##^
*p* < 0.01 vs. α-MSH-treated group or ET-1-treated group).

**Figure 5 ijms-24-07471-f005:**
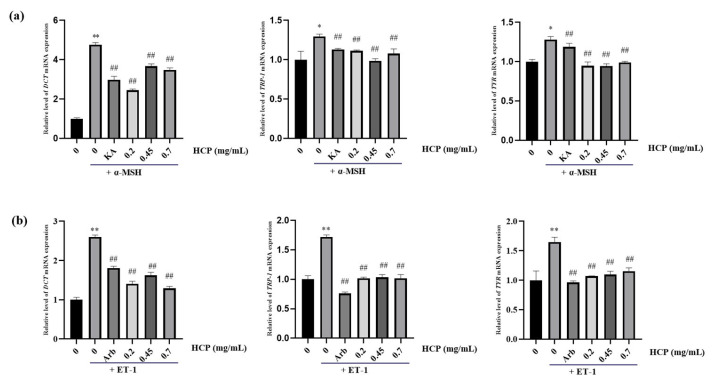
Effect of HCP on mRNA expression of tyrosinase (*TYR*), tyrosinase related protein 1 (*TRP-1*) and dopachrome tautomerase (*DCT*). The cDNA for *β*-*actin* served as the internal control. (**a**) Expression levels of melanogenesis-related genes, compared with *α*-MSH group. (**b**) Expression levels of melanogenesis-related genes, compared with ET-1 group. All data are expressed as mean ± SD (*n* ≥ 3) (** *p* < 0.01, and * *p* < 0.05 vs. the untreated control group; ^##^
*p* < 0.01 vs. *α*-MSH-treated group or ET-1-treated group).

**Figure 6 ijms-24-07471-f006:**
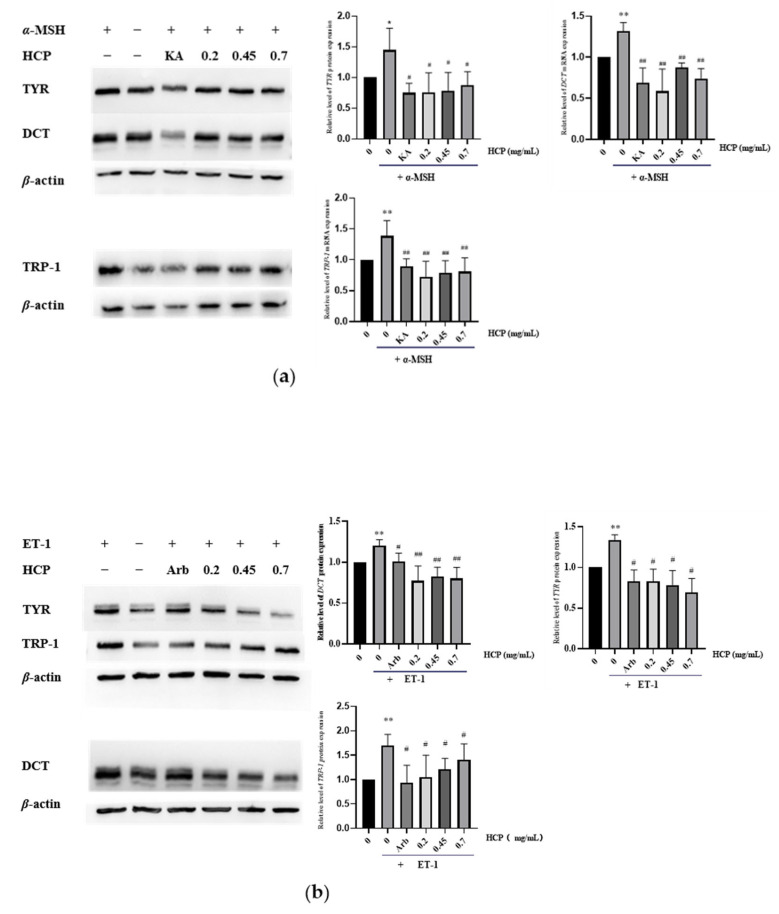
Effect of HCP on protein expression of TYR, TRP-1 and DCT. *β*-actin served as internal control. (**a**) Expression levels of melanogenesis-related protein, compared with *α*-MSH group. (**b**) Expression levels of melanogenesis-related genes, compared with ET-1 group. All data are expressed as mean ± SEM (*n* ≥ 3) (** *p* < 0.01, and * *p* < 0.05 vs. the untreated control group; ^##^
*p* < 0.01, and ^#^
*p* < 0.05 vs. *α*-MSH-treated group or ET-1-treated group).

**Figure 7 ijms-24-07471-f007:**
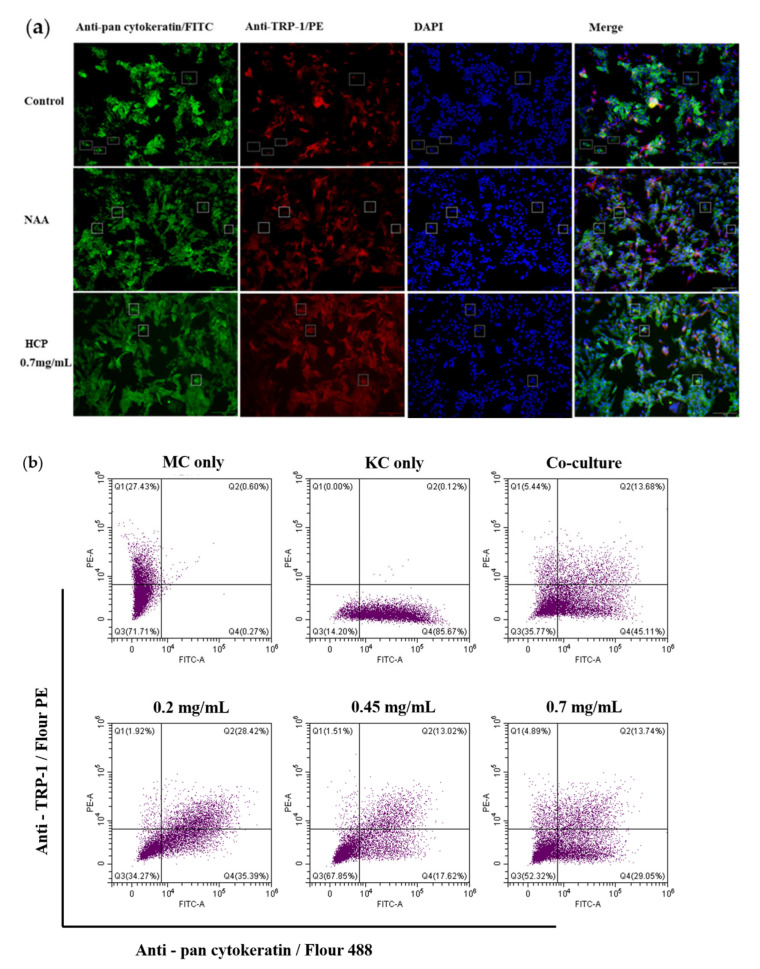
Effects on melanosome transfer in in the co-culture of melanoma and keratinocytes. Promotion of melanosome transfer in the co-culture of melanoma and keratinocytes in the presence and absence of HCP. (**a**) immunofluorescence staining; (**b**) flow cytometry. All data are expressed as mean ± SD (*n* ≥ 3) (*** *p* < 0.001, ** *p* < 0.01 and * *p* < 0.05 vs. the untreated control.)

**Table 1 ijms-24-07471-t001:** RT-qPCR prime.

Gene	Sequence
*β-actin*	Forward 5′-TTCTACAATGAGCTGCGTGTGG-3′
	Forward 5′-GTGTTGAAGGTCTCAAACATGAT-3′
*TYR*	Forward 5′-GCAAAGCATACCATCAGCTCA-3′
	Reverse 5′-GCAGTGCATCCATTGACACAT-3′
*TRP-1*	Forward 5′-TCTCTGGGCTGTATCTTCTTCC-3′
	Reverse 5′-GTCTGGGCAACACATACCACT-3′
*DCT*	Forward 5′-AACTGCGAGCGGAAGAAACC-3′
	Reverse 5′-CGTAGTCGGGGTGTACTCTCT-3′

## Data Availability

The data that support the findings of this study are available from the corresponding author upon reasonable request.

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
