# Peer review of "Hydrolyzed Conchiolin Protein (HCP) Extracted from Pearls Antagonizes both ET-1 and α-MSH for Skin Whitening"

_ijms, 2023, doi:10.3390/ijms24087471_

Round 1
Reviewer 1 Report
Authors are exploring the effect of Hydrolized conchiolin protein from pearl powder on melanogenesis.
The investigation is conducted by the use of relevant methodologies for in vitro investigations. Results obtained in the manuscript contribute to the overall pull of information which are giving scientific explanations for effects of substances used traditionally for centuries. In this way we can speak about evidence based actives in cosmetology. Relevance and specificity are average but this paper certainly adds to till now non-existent data.
Presented methodologies are satisfactory and conclusions are consistent with the evidence and arguments presented, figures in the paper are with adequate captions and informative.
References are appropriate, up to date, indicating interest in the area of skin whitening products and actives in cosmetology and dermatology.
Reviewer 2 Report
The paper is interesting. The studies are complete but before publishing, some small of the editorial errors should be removed:
- there is not explanation of * sign in the caption of figure 1
- in the figure 3 caption are mentioned signs (* and #) which are not shown in the figure 3
- There is repeated "in" preposition in the title of 2.7. chapter.
Reviewer 3 Report
It is a well written manuscript, with good presented results and enough good scientific soundness. Nevertheless, I would propose to enhance, either in discussion, or in conclusion the final interest regarding the non-adverse events of Hydrolyzed Conchiolin Protein (HCP) from pearl powder versus arbutin, kojic acid and especially hydroquinone, which may cause a carcinogenic effect and is prohibited in EU, USA etc. It is the final benefit of pearl component in cosmetic science.
Hear are some minor corrections:
Lines 78,81, 109, 115 : You write either nmol/L for concentrations or mg/mL. You should use the same measurement unit, especially when you compare between the parameters in the same paragraph, like 2.3
Line 197: Probably you mean co-culture instead coculture.
